# The Conserved Transcriptional Activation Activity Identified in Dual-Specificity Tyrosine-(Y)-Phosphorylation-Regulated Kinase 1

**DOI:** 10.3390/biom13020283

**Published:** 2023-02-02

**Authors:** Xiuke Ouyang, Zhuqing Wang, Bingtong Wu, Xiuxia Yang, Bo Dong

**Affiliations:** 1Fang Zongxi Center, MoE Key Laboratory of Marine Genetics and Breeding, College of Marine Life Sciences, Ocean University of China, Qingdao 266003, China; 2Laoshan Laboratory, Qingdao 266237, China; 3Institute of Evolution & Marine Biodiversity, Ocean University of China, Qingdao 266003, China

**Keywords:** DYRK1, transcriptional activation, ion transport, neuronal function, intellectual disability syndrome

## Abstract

Dual-specificity tyrosine-(Y)-phosphorylation-regulated kinase 1 (DYRK1) encodes a conserved protein kinase that is indispensable to neuron development. However, whether DYRK1 possesses additional functions apart from kinase function remains poorly understood. In this study, we firstly demonstrated that the C-terminal of ascidian *Ciona robusta* DYRK1 (CrDYRK1) showed transcriptional activation activity independent of its kinase function. The transcriptional activation activity of CrDYRK1 could be autoinhibited by a repression domain in the N-terminal. More excitingly, both activation and repression domains were retained in HsDYRK1A in humans. The genes, activated by the activation domain of HsDYRK1A, are mainly involved in ion transport and neuroactive ligand–receptor interaction. We further found that numerous mutation sites relevant to the *DYRK1A*-related intellectual disability syndrome locate in the C-terminal of HsDYRK1A. Then, we identified several specific DNA motifs in the transcriptional regulation region of those activated genes. Taken together, we identified a conserved transcription activation domain in DYRK1 in urochordates and vertebrates. The activation is independent of the kinase activity of DYRK1 and can be repressed by its own N-terminal. Transcriptome and mutation data indicate that the transcriptional activation ability of HsDYRK1A is potentially involved in synaptic transmission and neuronal function related to the intellectual disability syndrome.

## 1. Introduction

Dual specificity tyrosine-(Y)-phosphorylation-regulated kinase (DYRK), as a protein kinase, plays vital regulatory roles in development and disease [1,2,3,4]. There are five DYRK family members in mammals: DYRK1A, DYRK1B, DYRK2, DYRK3, and DYRK4 [5], among which DYRK1A plays exclusive roles in neurological diseases [1,4,6,7]. *DYRK1A* has been reported to be constitutively expressed at a high level in the central nervous system (CNS) of *Rattus* [8]. In chick and mouse embryos, *DYRK1A* is detected in neuronal progenitors and differentiated neurons [9,10]. DYRK1A distributes in different types of neuronal cells in the human brain [11] and has been found to be associated with neurological diseases, such as Down syndrome (DS), autism, and neurodegenerative diseases [12]. DS is the most common human genetic disorder and the major cause of mental retardation. Most DS patients have typical symptoms, including cognitive defects and memory impairment [1,13]. The main pathogenic factor of DS is the trisomy of chromosome 21, and *DYRK1A* locates on the DS critical region (DSCR) of chromosome 21. Therefore, the extra copy increases *DYRK1A* expression level by approximately 1.5-fold in DS brains [6]. Overexpression of *DYRK1A* has been demonstrated to cause morphological and cognitive defects in mouse DS models [12]. Additionally, it is well known that DS patients present a size-reduced brain [14]. Previous studies showed that *DYRK1A* regulates brain size in a dose dependent manner in mouse, *Xenopus*, and *Drosophila* [15,16,17], indicating that the expression of *DYRK1* might be regulated rigidly for maintaining the normal development and function of brain. However, the molecular mechanisms of how *DYRK1A* regulates the downstream genes remains elusive.

DYRK1 phosphorylates a broad set of proteins involved in diverse cellular processes, such as cell cycle, cell differentiation, and cell apoptosis [18,19,20]. DYRK1A phosphorylates nuclear factor of the activated T-cell (NFAT) to induce its nuclear export [21]. DYRK1 regulates the transcriptional activity of Gli1 by promoting its nucleus localization [22]. DYRK1A prevents the proliferation of embryonic neuronal cells by phosphorylating p53 at Ser15 to enhance p53 transcriptional activity [23]. One finding also indicates that *DYRK1A* participates in transcriptional regulation. DYRK1A shows transcriptional activation activity that is dependent on kinase activity of itself [24]. *DYRK1A* could increase *Tau* mRNA stability in a kinase-independent manner, thereby enhancing the expression level of *Tau* [25]. These findings suggest that *DYRK1* regulates the gene expression through some new mechanisms. Whether DYRK1 possesses an authentic transcription activation domain independent of the kinase domain remains unrevealed.

As a branch of chordate, ascidian *Ciona*’s genome does not duplicate and remains much ancestral single-copy gene [26,27]. It is a suitable ancestor material for the research of conservation or divergency of gene function. In this study, we identified a DYRK homology CrDRYK1 in *Ciona* and proved that the C-terminal of CrDYRK1 possessed the transcriptional activation activity that was regulated by the repression domain localizing in the N-terminal. We then demonstrated that the transcriptional activation and repression mechanism were retained in *HsDYRK1A* but not in *HsDYRK1B*. Using RNA-seq, we further revealed that the transcriptional activation ability of HsDYRK1A was mainly involved in ion transport and neuroactive ligand–receptor interaction. Taken together, we demonstrated that, apart from kinase activity, CrDYRK1 possessed transcriptional activation activity independent of kinase activity and regulated by the repression domain of itself. The on–off mechanism of transcriptional activation activity was conserved in HsDYRK1A, and the transcriptional activation ability might be responsible for neuronal development and function.

## 2. Materials and Methods

### 2.1. Animals and Electroporation

Animals were acquired from the eastern coast of Qingdao (Shandong, China). Mature eggs and sperm were collected and fertilized in seawater at room temperature; dechorionation and electroporation were performed as previously described [28]. After electroporation, dechorionated eggs were washed and cultured at 16 °C.

### 2.2. Plasmid Constructions

PCR was performed with the Phanta Max Super-Fidelity DNA Polymerase (Vazyme, Nanjing, China). DNA was purified using a GeneJET Gel Extraction Kit (Thermofisher, Scientific, Waltham, MA, USA). All DNA Fragments were ligated by the One Step Cloning Kit (Vazyme, Nanjing, China).

To identify the transcriptional activation activity, the isoform I, isoform II, and truncated isoform I of *CrDYRK1* were amplified from *C. robusta* cDNA separately using PCR and ligated to pGBTK7 (BD) vector (Takara, Beijing, China), respectively. The *HsDYRK1A* (residues 485–763) and *HsDYRK1B* (residues 437–629) fragments were amplified separately from human cDNA by PCR and ligated to BD vector for the transcriptional activation activity analysis. The *CsDYRK1* (residues 393–626) fragment was amplified from *C. savignyi* cDNA by PCR and ligated to BD vector for the transcriptional activation activity analysis. The *DmMNB* (residues 629–1047) fragment was amplified from *Drosophila melanogaster* cDNA by PCR and ligated to BD vector for the transcriptional activation activity analysis.

In the repression assay, *CrNFAT5* (residues 77–979) and *CrMyc* (residues 1–681) fragments were amplified and ligated to BD vector to serve as a positive control. *CrDYRK1* (isoform II) (residues 1–168) fragment was amplified and subcloned into EcoRI restriction sites of the CrNFAT5 (residues 77–979)-BD vector. *CrDYRK1* (isoform II) (residues 1–168) and *CrMyc* (residues 1–681) fragments were amplified separately and ligated to BD vector to construct an expression vector CrDYRK1 (isoform II) (residues 1–168)-CrMyc (residues 1–681)-BD. Truncated versions of *CrDYRK1* (isoform II) were amplified separately and ligated to BD vector to construct expression vector CrDYRK1 (isoform II) (residues 43–815)-BD and CrDYRK1 (isoform II) (residues 85–815)-BD. *HsDYRK1A*-N (1–69) fragment was amplified and subcloned into EcoRI restriction sites of the HsDYRK1A (residues 485–763)-BD vector to construct expression vector HsDYRK1A-N (residues 1–69)-C (residues 485–763)-BD.

In the RNA-seq assay, *HsDYRK1A* (residues 485–763) fragment was amplified and ligated to pEGFP-N1 vector (Addgene, Watertown, MA, USA). *HsDYRK1A*-N (1–69)-C (485–763) was amplified from HsDYRK1A-N (residues 1–69)-C (residues 485–763)-BD vector and ligated to pEGFP-N1 vector.

In the assay of subcellular localization observation, the isoform I and isoform II of *CrDYRK1* were amplified and ligated to pEGFP-C1 vector driven by *Brachyury* promoter. *HsDYRK1A* (residues 485–763) fragment was amplified by PCR and ligated to pEGFP-N1 vector. *CsDYRK1* (residues 393–626) and *CrDYRK1* (isoform I) (residues 427–647) fragments were amplified by PCR and ligated to pEGFP-N1 vector driven by *Epi I* promoter, respectively.

All primers are listed in Appendix A.

### 2.3. Gene Structure, Phylogenetic Analysis, Domain Analysis

The genomic sequence and coding sequence (CDS) of *CrDYRK1* were downloaded from the Ghost Database (http://ghost.zool.kyoto-u.ac.jp/cgi-bin/gb2/gbrowse/kh/) (accessed on 10 October 2019) and used for analyzing gene structure in the GSDS-2.0 (http://gsds.gao-lab.org/) (accessed on 10 October 2019). The amino acids sequence of the DYRK1 from different species were acquired from the NCBI, and the sequences of kinase domains were used to build phylogenetic tree by MEGA-X software. The phylogenetic tree was built using the maximum likelihood method with 1000 bootstraps, and the JTT model was selected. The protein 3D structure was predicted using online tools ALPHAFOLD (https://cosmic-cryoem.org/tools/alphafold/). The conserved domain analysis of DYRK1 was performed by NCBI CD-Search (https://www.ncbi.nlm.nih.gov/Structure/cdd/wrpsb.cgi) (accessed on 18 December 2021) and SMART (http://smart.embl-heidelberg.de/) (accessed on 18 December 2021). To reveal the preference of *HsDYRK1A*-activating genes, DNA motif analysis was performed using online tool MEME (https://meme-suite.org/meme/doc/meme.html) (accessed on 19 November 2022). The upstream 1000 bp of translation start sites of 96 upregulated genes were obtained for motif analysis.

### 2.4. Sequence Alignment and Ks Calculation

The alignment of the amino acids sequence from different species was performed by using DNAMAN software. The coding sequence of the *DYRK1* from different species were collected from the NCBI. To examine whether these duplicated *DYRK1* members diversified, synonymous (Ks) substitution of each duplicated *DYRK1* were calculated using KaKs_Calculator 2.0 (KaKs_Calculator2.0 download | SourceForge.net) [29].

### 2.5. Yeast Two-Hybrid Assay

The vectors of BD and pGADT7 (AD), the Y2H Gold strain, and all reagents for Y2H assay were purchased from Takara. Different combinations of bait with AD empty vectors were co-transformed into the yeast strain Y2H Gold and cultured on synthetic complete medium lacking leucine and tryptophan (–Leu, –Trp). The transcription activation ability was determined by measuring the growth of serial dilutions on synthetic complete medium lacking leucine and tryptophan synthetic supplemented with Aureobasidin A (AbA) for 2 to 3 d.

### 2.6. Cell Culture, RNA Sequencing, Data Analysis, and MOTIF Enrichment Analysis

HeLa cells were cultured in DMEM supplemented with 10% fetal bovine serum (Thermofisher Scientific, Waltham, MA, USA) at 37 °C with 5% CO2. The plasmids were transfected into cultured cells according to the manufacturer’s instructions (Lipo3000 from Thermofisher).

Sixty hours post-transfection, we extracted the total RNA from HeLa cells overexpressing HsDYRK1A (residues 485–763)-eGFP or HsDYRK1A (residues 1–69, 485–763)-eGFP, respectively. Transcriptome sequencing was performed by the Berry Genomics company. Genes were annotated according to genome Homo sapiens-NCBI-hg19 (ftp://ftp.ncbi.nlm.nih.gov/genomes/all/GCF/000/001/405/GCF_000001405.25_GRCh37.p13/GCF_000001405.25_GRCh37.p13_genomic.fna.gz) (accessed on 25 October 2022). EdgeR was used for differential expression analysis. The resulting *p*-values were adjusted using Benjamini and Hochberg’s approach for controlling the false discovery rate. Genes with log2 (fold change) > 1 and *p* value < 0.05 were assigned as differentially expressed. GO and KEGG enrichment analysis of differentially expressed gene sets were implemented using the topG (http://www.bioconductor.org/packages/release/bioc/html/topGO.html) (accessed on 9 November 2022) and KOBAS package, respectively.

### 2.7. Immunostaining and Imaging

The collected embryos were fixed with 4% paraformaldehyde (Sigma Aldrich, St. Louis, MO, USA) at room temperature for 2 h and then were washed thrice with phosphate-buffered saline (PBS) containing 0.1% Triton X-100 (Solarbio, Beijing, China) (PBST) for 20 min each time. To identify the cell boundary, fixed embryos were stained with 1/200 TRITC phalloidin (YEASEN, Shanghai, China) in PBST overnight at 4 °C. Embryos were then washed thrice with PBST and mounted using Vectashield mounting medium with 4′,6-diamidino-2-phenylindole (DAPI) (Vector Laboratories, Newark, CA, USA) to label the cell nucleus.

HeLa cells were fixed with 4% formaldehyde for 1 h at RT. The fixed cells were rinsed thrice with PBST for 10 min each time. To identify the cell boundary, fixed cells were stained with 1/500 TRITC phalloidin in PBST overnight at 4 °C. HeLa cells were then washed thrice with PBST and mounted using Vectashield mounting medium with DAPI to label the cell nucleus.

Images were taken using a Zeiss LSM900 Confocal Microscope (Zeiss, Oberkochen, Germany).

## 3. Results

### 3.1. Conservation of Kinase Domain in DYRK1 in the Evolution

To better understand the origin and evolution of DYRK1, phylogenetic tree was constructed based on the conserved kinase domain. The results showed that two homologs of DYRK1, DYRK1A and DYRK1B, were found in most vertebrates, such as *Danio rerio*, *Mus musculus*, and *Homo sapiens*. However, only one homolog of DYRK1 was identified in Urochordata ascidian *Ciona*, suggesting that DYRK1 gene underwent duplication during the evolution (Figure 1A). To identify the functional conservation of DYRK1, we performed the domain analysis and found that the kinase domain was distributed in all DYRK1, indicating the conserved kinase function. In addition, DYRK1A, distinct from DYRK1B, possessed a polyhistidine stretch that is important for nuclear speckles localization [30] (Figure 1B), suggesting the functional difference between DYRK1A and DYRK1B. Alignment of amino acids of kinase domain analysis showed that the sequence identities of DYRK1A proteins were higher than those of DYRK1B proteins (Figure 1C). We further calculated the synonymous substitution rate (Ks) of DYRK1A and DYRK1B based on CrDYRK1 by using KaKs_Calculator 2.0 [29]. The results showed that Ks of DYRK1A was higher than DYRK1B (Figure 1D), suggesting DYRK1A to be the original copy. Collectively, these results suggest that the kinase function of DYRK1 is highly conserved during the evolution, of which DYRK1A might be ancestral copy.

### 3.2. CrDYRK1 Possesses Transcriptional Activation Ability

There are three isoforms of CrDYRK1 in ascidian *C. robusta*. Isoform II and isoform III shared same coding sequence, which was 168 coding amino acids longer than isoform I at the N-terminal (Figure 2A). To examine whether *CrDYRK1* has a transcriptional activation function, the transcriptional activation activity verification was performed using a yeast assay system. Yeast strain Y2H GOLD contains reporter gene *AUR1-C* that is the dominant mutant version of *AUR1* (aureobasidin resistance) and endows the resistance of Y2H for AbA. The AD and BD vectors contain the encode sequence of leucine (Leu) and tryptophan (Trp), respectively. Then, the coding sequences of isoform I and isoform II were amplified from *C. robusta* and fused to the GAL4 DNA-binding domain to generate CrDYRK1 (isoform I)-BD and CrDYRK1 (isoform II)-BD fusion plasmids. The plasmids were co-transformed into the yeast strain Y2H with an AD empty vector, respectively. The transformants were evaluated for their abilities to activate reporter gene *AUR1-C* transcription. All transformants could grow on the medium lacking leucine and tryptophan, indicating that AD and BD were transformed into Y2H strains. Excitingly, the strains transformed with CrDYRK1 (isoform I) grew on medium supplemented with AbA but not the strains transformed with CrDYRK1 (isoform II) (Figure 2B), suggesting that CrDYRK1 (isoform I) possessed transcriptional activation activity. Furthermore, we constructed different truncated versions of CrDYRK1 (isoform I) to examine the exact activation domain. We found that both fragment (residues 1–87) and fragment (residues 88–426) of CrDYRK1 (isoform I) did not show the transcriptional activation activity, whereas the fragment (residues 427–647) had this activity (Figure 2C). Collectively, these results indicated that CrDYRK1 possessed transcriptional activation ability, and the activation domain located in the C-terminal that was separate from the kinase domain.

### 3.3. Identification of an Active Repression Domain in CrDYRK1 (Isoform II)

The activation domain was present in both isoform I and isoform II of CrDYRK1. However, only isoform I possessed transcription activation ability. Based on this result, we speculated that CrDYRK1 (isoform II) (residues 1–168) contained a repression domain to inhibit the transcriptional activation activity. To verify this idea, CrDYRK1 (isoform II) (residues 1–168) was fused with two transcription factor CrNFAT5 and CrMyc for function analysis, respectively. The results showed that the transcriptional activation activity of CrNFAT5 or CrMyc was presented; however, their transcriptional activation activities were obviously suppressed after fused with CrDYRK1 (isoform II) (residues 1–168), which indicates that residues 1–168 of CrDYRK1 (isoform II) functioned as a repression domain (Figure 2D,E). Then, the truncated versions were used for exploring key region of the repression domain in CrDYRK1. We found that deletion of residues 1–42 did not affect the suppression effect, while the transcriptional activation activity appeared after deletion of residues 1–84 (Figure 2F), indicating that residues 1–84 of CrDYRK1 are responsible for repression function. Next, protein 3D structure analysis revealed that the repression domain and activation domain of CrDYRK1 (isoform II) did not come into physical contact, which implies that the suppression effect from the repression domain might not rely on the intramolecular physically binding between the repression domain and activation domain (Appendix A). Then, we expressed eGFP-CrDYRK1 (isoform I) and eGFP-CrDYRK1 (isoform II) in *Ciona* notochord cells driven by the *Brachyury* promoter. The results showed the different subcellular localization of CrDYRK1 with and without the repression domain. The GFP signal was detected at the nucleus in the isoform I-expressing cells (Appendix A). Meanwhile, the GFP signal was detected at the nucleus, cytoplasm, and cell boundary in the isoform II-expressing cells (Appendix A). Collectively, our results demonstrated the existence of a repression domain in the N-terminal that could inhibit the transcriptional activation activity of the C-terminal in CrDYRK1 (isoform II).

### 3.4. The Conservation of the Transcriptional Activation Ability in Human DYRK1

To identify the conservation of transcriptional activation, the homologous regions with a CrDYRK1 (isoform I) activation domain (residues 427–647) from different species were amplified and used for function analysis in yeast system, respectively (Appendix A). The results showed that human HsDYRK1A (residues 485–763) presented the transcriptional activation activity but HsDYRK1B (residues 437–629) did not. CrDYRK1 (isoform I) (residues 427–647) and another ascidian species *C. savignyi* CsDYRK1 (residues 393–626) also showed the transcriptional activation activity. However, the homologous region (residues 629–1047) of DYRK1A in *Drosophila* (also called MNB) did not show the transcriptional activation activity either (Figure 3A). Then, we found that N-terminal (residues 1–68) of HsDYRK1A also contained a repression domain that could inhibit the transcriptional activation activity of the C-terminal (residues 485–763) (Figure 3B). Collectively, the transcriptional activation ability of DYRK1 was conserved in Urochordate and humans, despite the divergency of HsDYRK1B from HsDYRK1A.

### 3.5. The Function Analysis of HsDYRK1 Transcription Activation Ability

Nuclear localization is a prerequisite for proteins to perform the transcriptional activation function. To observe subcellular localization, the HsDYRK1A (residues 485–763) fused with eGFP was expressed in HeLa cells, and the GFP signal was detected at the nucleus in the HsDYRK1A (residues 485–763)-expressing cells (Appendix A). Then, we expressed CsDYRK1 (residues 393–626)-eGFP and CrDYRK1 (isoform I) (residues 427–647)-eGFP in *Ciona* epidermis cells driven by *Epi I* promoter. The GFP signal was detected obviously at the nucleus in both the CsDYRK1 (residues 393–626)-expressing cells and CrDYRK1 (isoform I) (residues 427–647)-expressing cells (Appendix A). These results indicated that the activation domain of DYRK1 might also localize in cell nucleus despite the absence of two nuclear localization signals of the N-terminal.

To further reveal the functions of the HsDYRK1’s transcriptional activation ability, the RNA-seq assay was applied to identify its downstream genes. The truncated HsDYRK1A with a transcription activity domain from residues 485 to 763 was overexpressed in HeLa cells, and the truncated HsDYRK1A from residues 1 to 68 plus 485 to 763 was overexpressed as the control. Transcriptome assay identified that 235 differential expression genes (DEGs) (fold change >2 or <0.5 and *p*-value < 0.05) including 105 upregulated genes and 130 downregulated ones (Figure 4A, Appendix A). Most DEGs were predicted to localize in the plasma membrane and cell periphery using cellular component (CC) analysis (Figure 4B). For those upregulated genes, Gene Ontology (GO) enrichment analysis revealed that they were mainly involved in regulation of ion transport, especially for calcium ions, in the biological process (BP) (Figure 4C), and were enriched in calmodulin binding, motor activity, and sodium ion transmembrane transporter activity in molecular function (MF) (Figure 4D). The Kyoto Encyclopedia of Genes and Genomes (KEGG) analysis showed that the upregulated genes were related to neuroactive ligand–receptor interaction and cytokine–cytokine receptor interaction (Figure 4E). Furthermore, the same analysis procedures were applied to the downregulated genes. The results showed that they were mainly implicated in kidney development in BP analysis and enriched in ion antiporter activity in MF analysis (Appendix A). The KEGG results showed that they were also mainly related to neuroactive ligand–receptor interaction and cytokine–cytokine receptor interaction (Figure 4F). Some DEGs (such as *ANK2*, *ATP1B2*, *OPRL1*, *PDE4B*, *TTYH1*, *ENKUR*, *PNCK*, *SLC8A2*, *GRIN3A*, *GPR83*, and *GRHR2*) involved in ion transport or neuroactive ligand–receptor interaction were highly expressed in the brain (Appendix A). Proper ion transport is vital to synaptic transmission and neuronal development [31,32], and the transmission of signal relies on some important receptor proteins in neuron, for example, GRIN3A and GPR83.

We further investigated the correlation between amino acids mutation of the activation domain and human disease in the UniProt database (https://www.uniprot.org/uniprotkb/Q13627/entry). The result showed that there were many mutations locating in the HsDYRK1A activation domain that might be involved in DYRK1A-related intellectual disability syndrome (Figure 5).

Taken together, these results indicated that the transcriptional activation ability of HsDYRK1A was mainly implicated in ion transport and neuroactive ligand–receptor interactions that were crucial for neuronal function performance and potentially related to neuron diseases.

### 3.6. The Motif Analysis Reveals the Preference of the Activation Domain of HsDYRK1

To investigate the mechanism of transcription activation in HsDYRK1, we performed motif analysis to find out the particular DNA sequences that HsDYRK1 potentially bound to using the online tool MEME (https://meme-suite.org/meme/doc/meme.html). Five most representative motifs were identified from the upstream 1000 bp of translation start site of 96 upregulated genes (Figure 6A). There were 14 promoters that contained all five representative motifs. The adenine-rich motif-1 (RADACTCYRTYTYAAAAAAAAAAAAAAAAAWAAARAAAAAW) has a mostly broad distribution up of to 73.95% (71/96). However, the number of motif occurrence was no more than 22 in the remaining four motifs, suggesting the obvious preference of HsDYRK1A (Figure 6B). These enriched motifs were mainly located at the upstream 500–1000 bp of the translation initiation site (Figure 6C). These results suggest that the activation domain of HsDYRK1A prefers to activate genes that contain specific DNA motifs.

## 4. Discussion

*DYRK1A* plays vital roles in embryogenesis and organogenesis, which has been implicated in pathogenesis of neuronal diseases [12,16,33]. The kinase domain of DYRK1 is conserved in most of the species and its ability to phosphorylate downstream proteins is essential to DYRK1 functioning [5]. *Ciona* belongs to the class Ascidiacea, subphylum Urochordata, and phylum Chordata. Urochordates are the invertebrates most closely related to vertebrates [34]. The ascidian *Ciona* is an important model organism for developmental and evolutionary biology [35]. Given that *Ciona* has nonwhole genome duplication and remains many ancestral single-copy genes, it is regarded as a suitable ancestral material for studying the conservation or divergency of gene function [26,27]. In this study, we demonstrated that the transcriptional activation ability presented in the C-terminal of *Ciona* CrDYRK1, which was also retained in human HsDYRK1A. These results suggest that DYRK1 possessed the transcriptional activation ability apart from kinase function, and this ability might be conserved in chordates. Our result is supported by previous results. Previous research demonstrated that the full length DYRK1A shows transcriptional activation activity [24], and that kinase activity is indispensable to DYRK1A transcriptional activation activity [24]. This research, in agreement with our finding, suggested that DYRK1A possesses potential transcriptional activation ability. However, the dependence of transcriptional activation activity on kinase activity remains controversial. Our findings presented that the activation activity of DYRK1A is independent of its kinase function. The activation domain, located in the C-terminal of DYRK1A, individually performed the activation function. In addition, another study showed that full-length DYRK1A is able to enhance the *Tau* expression by increasing the mRNA stability, and the C-terminal region of DYRK1A plays a vital role in this process [25]. The finding revealed an additional function of *DYRK1A* that is distinct from our research. It indicates that *DYRK1A* could regulate gene expression using different approaches, which enriches our understanding of *DYRK1A* function. The research also demonstrates the importance of the C-terminal region for DYRK1 function [25]. We did not identify a classical DNA binding domain in full-length HsDYRK1A. However, the motif analysis demonstrated that *HsDYRK1A* had an obvious preference for target genes containing specific DNA motifs. Thus, we speculate that a DNA binding protein might mediate the association of *DYRK1A* to the specific elements of DNA.

In this study, we found that the full-length CrDYRK1 (isoform I) rather than CrDYRK1 (isoform II) possessed transcriptional activation activity. CrDYRK1 (isoform II) is 168 amino acids longer than CrDYRK1 (isoform I) at the N-terminal. We further demonstrated that N-terminal (residues 1–168) of CrDYRK1 (isoform II) contained a repression domain, which could suppress the transcriptional activation activity. Then, we found that the on–off mechanism of transcriptional activation activity was retained in *HsDYRK1A*. The on–off mechanisms of transcriptional activation activity also exist in plant transcriptional factors. *NAC20*, a plant-specific transcription factor, is vital to multicellular processes. NAC20 shows highly transcriptional activation activity in the C-terminal. In the meantime, there is an active repression domain in the N-terminal of NAC20, which can inhibit transcriptional activation activity [36]. This indicates that the regulation pathway of transcriptional activation activity might be conserved in different organisms. In addition, the activation domain of HsDYRK1A is not distributed in all isoforms, suggesting that the HsDYRK1A executes a transcriptional activation function depending on the coordination of differential isoforms. Except for the repression function, the N-terminal of the 168 amino acids seems to also be responsible for the subcellular localization of DYRK1. In this research, the version of DYRK1 with transcriptional activation activity (isoform I) was shown to be enriched in the nucleus. The suppressed version of DYRK1 (isoform II) was mainly distributed in the cytoplasm and cell membrane, apart from the nucleus. The substrates of DYRK1A were distributed in various subcellular structures, including the nucleus, cytoplasm, cytoskeleton, and vesicles [37], which requires DYRK1 to distribute in different compartments of cell for phosphorylating diverse substrates. CrDYRK1 could achieve multiple localization by relying on different isoforms in *C. robusta*. Previous research also found that the DNA ligase 1 (Dlg1) or APL-like transcription factor has dual localization in mitochondria and the nucleus via alternative splicing, which generates different transcript-coding variant proteins in plants [38,39,40]. These findings suggest that there may be a universal mechanism that allows the encoded proteins of different transcripts from a certain gene to achieve distinct subcellular localization, thus exhibiting their own functions. The nuclear localization of CrDYRK1 (isoform I) is also more conducive to performing the transcriptional activation function, reflecting the precise regulation of function and localization. The N-terminal (residues 1–168) of CrDYRK1 (isoform II) might execute its suppression by changing the distribution of CrDYRK1 from the nucleus to the cytoplasm and cell membrane.

In this study, some of DEGs (for example, *ATP1B2*, *TTYH1*, *ENKUR*, *SLC8A1,* and *SLC8A2*), enriched by GO analysis, are highly expressed in the brain and crucial for ion transport [31,41,42,43,44], especially Ca^2+^ transmembrane transport. The communication between neurons is pivotal to cognition, including learning and memory. The process of synaptic transmission is regulated by ion transport [45]. Ca^2+^ signaling exists in all types of neurons [45], and the Ca^2+^ dyshomeostasis disrupts synaptic transmission [46]. DYRK1A-related neurological diseases show some clinical symptoms such as cognitive disorder with learning and memory impairment [1,13]. These symptoms might be caused by blockage of synaptic transmission. In addition, the MF analysis of transcriptome showed that the upregulated genes were enriched in motor activity. A previous finding indicates that the overexpression of *DYRK1A* caused the motor abnormalities in a murine model of DS [47]. Moreover, many nonsense or missense mutations that occurred in the activation domain of HsDYRK1A are associated with the DYRK1A-related intellectual disability syndrome. These evidences suggest that the transcriptional activation domain is a potential therapeutic target for DYRK1A-related neurological diseases apart from its kinase domain.

The KEGG analysis showed that DEGs were mainly enriched in neuroactive ligand–receptor interaction terms. Some of DEGs, related to neuroactive ligand–receptor interaction, are also highly expressed in the brain (for example, *OPRL1*, *GRIN3A*, *GPR83*, and *CRHR2*). The result also suggests that the transcriptional activation ability of HsDYRK1A was implicated in neuronal functions. Thus, the roles of *HsDYRK1A* in neuron development and function could be based on two distinct pathways: one is that HsDYRK1A phosphorylates downstream substrates relying on kinase function; another is that HsDYRK1A promotes gene expression depending on the transcriptional activation activity. Our finding provides novel understanding on the function of DYRK1A and suggests that the transcriptional activation domain is a potential target for the therapy of *DYRK1A*-related neurological diseases.

## Figures and Tables

**Figure 1 biomolecules-13-00283-f001:**
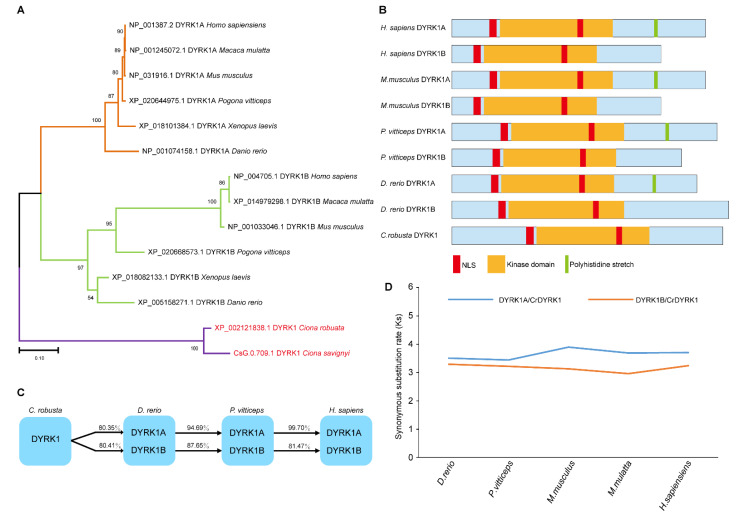
The protein kinase DYRK1 underwent gene duplication during evolution. (**A**) The phylogenetic tree analysis of DYRK1 from different species. A phylogenetic tree was constructed based on conserved kinase domain. There was a copy of *CrDYRK1* in *Ciona* and two copies, *DYRK1A* and *DYRK1B*, in most vertebrates. (**B**) The domain analysis of DYRK1. DYRK1 contained a highly conserved kinase domain (yellow) and two nuclear localization signals (red). A polyhistidine stretch (green) was only identified in DYRK1A. (**C**) The analysis of amino acid sequence identity. The amino acid sequence alignment was performed using the conserved kinase domain. The amino acid sequence identity of DYRK1A was higher than DYRK1B. (**D**) The synonymous substitution rate (Ks) of DYRK1. The Ks of DYRK1A and DYRK1B was calculated based on *C. robusta* CrDYRK1. The result showed that synonymous substitution rate (Ks) of DYRK1A was higher than that of DYRK1B.

**Figure 2 biomolecules-13-00283-f002:**
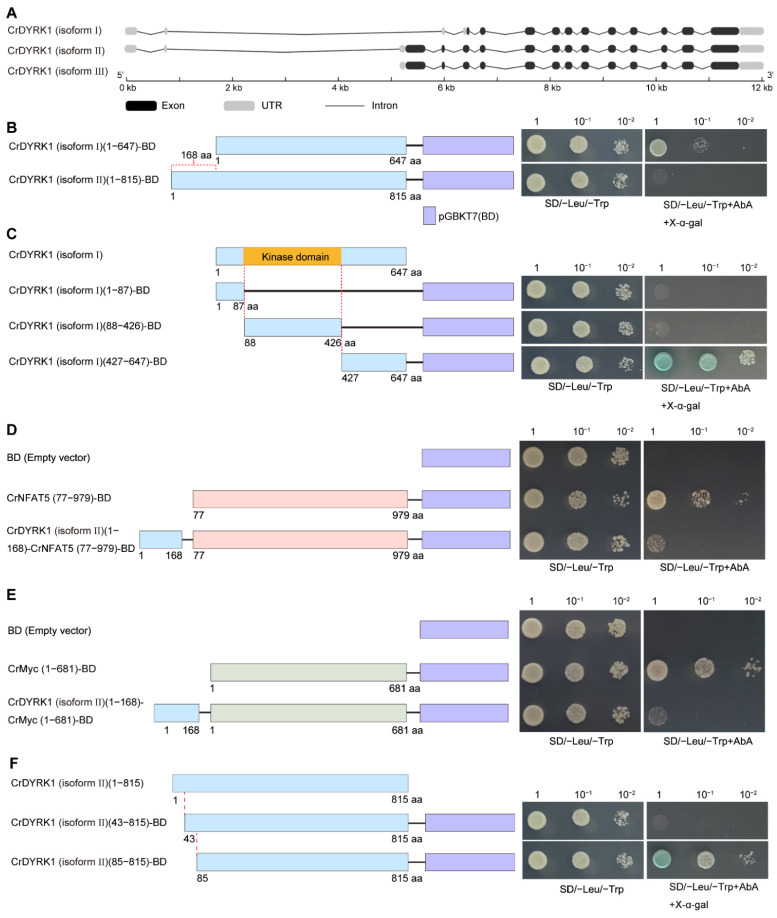
CrDYRK1 possessed an activation domain in C-terminal and a repression domain in N-terminal. (**A**) Schematic representation of different isoforms of CrDYRK1. The black box, grey box, and black line represent exon, untranslated regions (UTR), and intron, respectively. (**B**) CrDYRK1 possessed transcriptional activation activity. Isoform I and isoform II of CrDYRK1 were amplified from *C. robusta* and subcloned into BD vector with AD empty vector co-transformed into yeast strain Y2H, separately. Strain transformed with CrDYRK1 (isoform I) grew on medium supplemented with AbA (0.125 mg/L) and X-α-gal (40 mg/L). X-α-gal is another indicator of Y2H system. (**C**) The activation domain located in C-terminal of CrDYRK1. The truncate versions of CrDYRK1 (isoform I) were subcloned into BD vector for identifying the activation domain. The truncated CrDYRK1 (isoform I) (residues 427–647) showed transcriptional activation ability. And the activation activity was independent of kinase activity. (**D**,**E**) CrDYRK1 (isoform II) contained a repression domain in N-terminal (residues 1–168). CrDYRK1 (isoform II) (residues 1–168) was separately fused with two transcription factors, CrNFAT5 and CrMyc, for function analysis. The transcriptional activation activity of CrNFAT5 or CrMyc was obviously suppressed. (**F**) The residues 1–84 of CrDYRK1 (isoform II) are responsible for repression function. These truncated versions were used for exploring key region of CrDYRK1 repression domain, and the transcriptional activation activity appeared when deleting residues 1–84 of CrDYRK1 (isoform II). Left panel represent the vector construction and right panel indicate the yeast assay results in (**B**–**F**).

**Figure 3 biomolecules-13-00283-f003:**
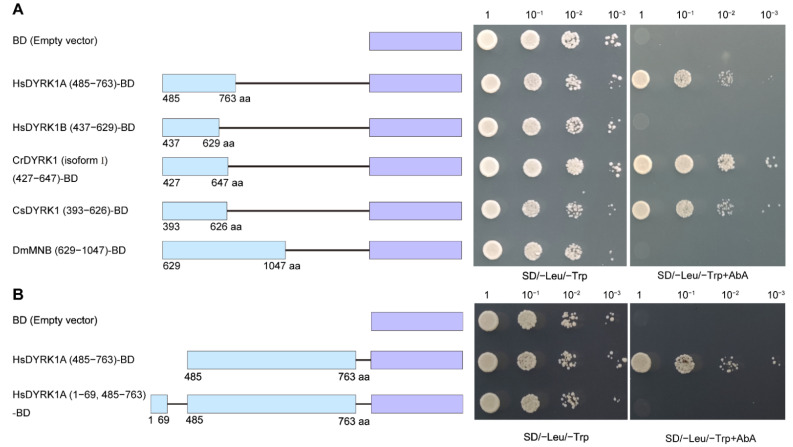
The on–off mechanism of transcriptional activation activity was conserved in HsDYRK1A. (**A**) The transcriptional activation ability was found in HsDYRK1A. The homologous regions of DYRK1 from diverse species with CrDYRK1 (isoform I) activation domain (residues 427–647) were amplified and used for function analysis in a yeast system, respectively. The result showed that human HsDYRK1A (residues 485–763) and another ascidian species *C. savignyi* CsDYRK1 (residues 393–626) presented transcriptional activation activity but HsDYRK1B (residues 437–629) did not. The homologous region (residues 629–1047) of DYRK1A in *Drosophila* (also called MNB) did not show the transcriptional activation activity either. (**B**) N-terminal of HsDYRK1A contained a repression domain. N-terminal (residues 1–69) could repress the transcriptional activation activity of HsDYRK1A (residues 485–763). Left panel represents the vector construction and right panel indicates the yeast assay results in (**A**,**B**).

**Figure 4 biomolecules-13-00283-f004:**
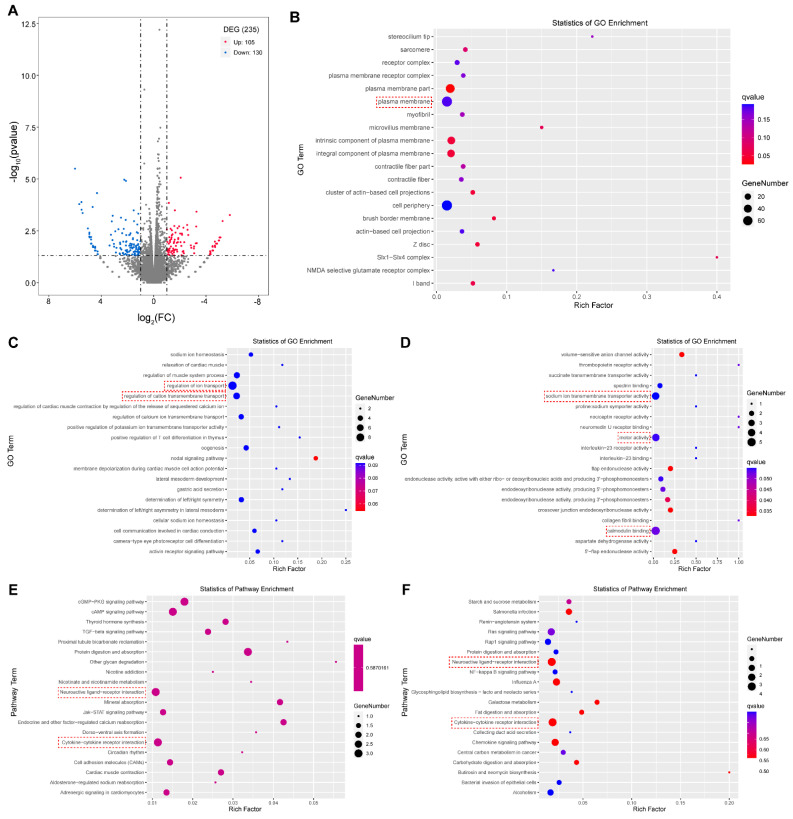
RNA-seq revealed potential roles of transactivation ability of HsDYRK1. (**A**) Volcano plot of the activation/repression group: 105 upregulated genes and 130 downregulated genes were identified (fold change >2 or <0.5 and *p*-value < 0.05). (**B**) Cellular component analysis of differential expression genes (DEGs): most DEGs were predicted to localize in plasma membrane (red box). (**C**) BP analysis of upregulated genes: upregulated genes were mainly involved in regulation of ion transport. (**D**) MF analysis of upregulated genes: upregulated genes were mainly enriched in calmodulin binding, motor activity, and sodium ion transmembrane transporter activity. (**E**) KEGG analysis of upregulated genes: upregulated genes were mainly enriched in neuroactive ligand–receptor interaction and cytokine–cytokine receptor interaction. (**F**) KEGG analysis of downregulated genes: downregulated genes were related to neuroactive ligand–receptor interaction and cytokine–cytokine receptor interaction.

**Figure 5 biomolecules-13-00283-f005:**
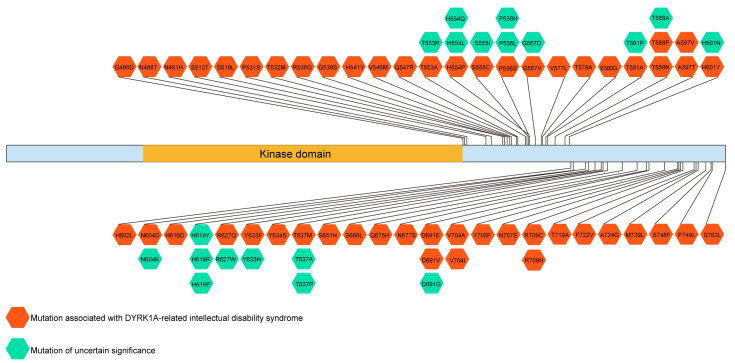
The activation domain of HsDYRK1A might be involved in DYRK1A-related intellectual disability syndrome. We collected the data from the UniProt database (https://www.uniprot.org/uniprotkb/Q13627/entry). Some mutations were correlated to DYRK1A-related intellectual disability syndrome. The red represents the mutation-associated DYRK1A-related intellectual disability syndrome. The green represents the mutation of uncertain significance.

**Figure 6 biomolecules-13-00283-f006:**
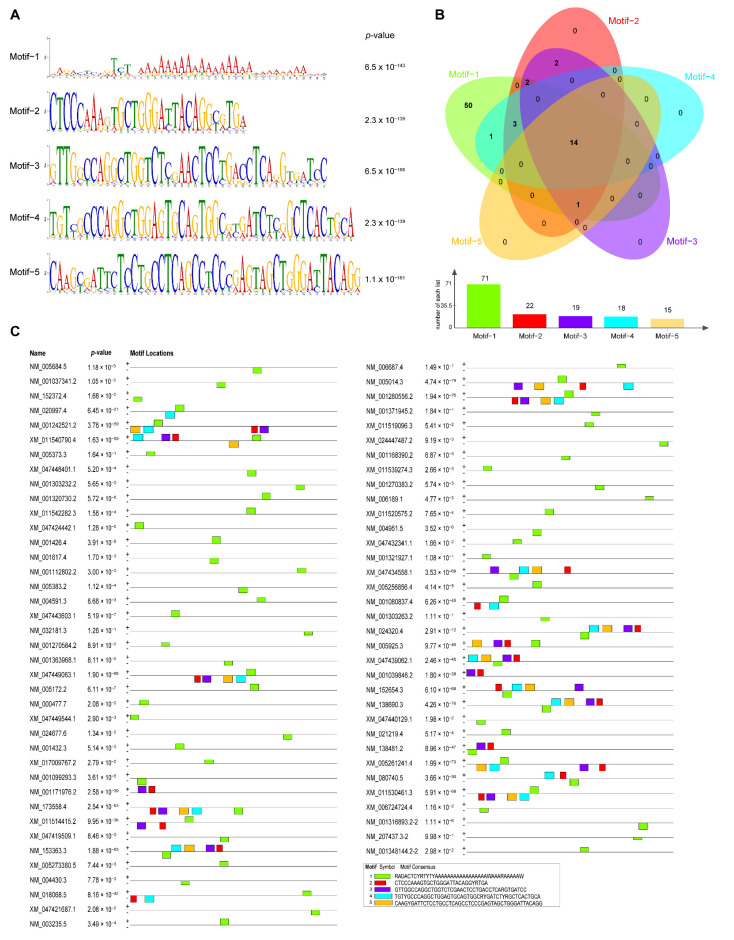
The motif analysis of upregulated gene promoters. (**A**) The representative DNA motifs of upregulated gene promoters: five representative motifs were shown. The height of the bases represents their frequency of occurrence at the positions. (**B**) The number of motif occurrences: There are 14 promoters that contained all five representative motifs. The adenine-rich DNA motif (RADACTCYRTYTYAAAAAAAAAAAAAAAAAWAAARAAAAAW) has the most broadly distributed at up to 73.95% (71/96). The Venn diagram was drawn using an online tool (https://bioinformatics.psb.ugent.be/webtools/Venn/). (C) The localization of motifs in the regulation region. These enriched motifs were mainly localized at the translation initiation site upstream 500–1000 bp of activated genes.

## Data Availability

The datasets supporting the conclusions of this article are included within the article and its additional files.

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
