# Peer review of "The Conserved Transcriptional Activation Activity Identified in Dual-Specificity Tyrosine-(Y)-Phosphorylation-Regulated Kinase 1"

_biomolecules, 2023, doi:10.3390/biom13020283_

Round 1
Reviewer 1 Report
Quyang et al., presented clear evidence in this manuscript to support the transcriptional function of CrDYRK1 that is independent of its kinase activity. The authors also identified the C-terminal activation domain and the N-terminal repressor domain related to the transcriptional function of CrDYRK1 and that this on-off mechanism is conserved in human DYRK1A but not DYRK1B. To explain the repression mechanism, the authors provided subcellular localization evidence to argue that the repressor domain enables extra nuclear localization of cDYRK1. However, a major concern is that only a small fraction of cytoplasmic and membranous localization cannot explain a complete loss of transcriptional activation for isoform II. Using RNA-seq and bioinformatics analyses, the paper provided potential transcriptional targets and pathways regulated by human hDYRK1A that is related to its role in neurological functions. Overall, the manuscript is well written. English editing is highly recommended to enhance readability. Before this paper is considered for publication, the following questions must be addressed by either experimentation or more insightful discussion.
Major Points:
1. A detailed description of the transcription activation assay must be inserted in section 3.2 to fully explain the data.
2. Supplemental Figure 1B must be inserted as a figure in Section 3.4 to support the point that residues of 1-168 of isoform II confers cytoplasmic and membranous localization. In this figure, co-localization of GFP-cDYRK1 with nuclear stain by DAPI and a membrane marker would significantly enhance the conclusion. Also, showing that GFP-cDYRK1 (isoform II) (85-815) loses its cytoplasmic and membranous localization is required to add further evidence to explain why this truncated protein acquire the ability for transcriptional activation, as shown in Fig. 2F. Is this also true for human DYRK1A? Is the presence of the N-terminal fragment 1-69 sufficient to relocate hDYRK1A (485-763) from the nucleus to the cytoplasm and membrane?
3. GFP-cDYRK1 (isoform II) still exhibited strong nuclear localization (supplemental figure 1b) yet it had little transcription activation activity, as compared with GFP-cDYRK1 (isoform I) (Fig. 2B). Therefore, it is impossible to explain the complete lack of transcription activity by only a small fraction of GFP-cDYRK1 (isoform II) outside the nucleus.
Minor Comments:
1. Line 255-258: “This indicated the suppression effect might not rely on the physically bound between repression domain and activation domain (Supplementary Figure 1A)”. The ALPHAFOLD prediction implicates that the suppression effect might not reply on intramolecular interaction between the repression domain and the activation domain. It is still possible the repression domain can bind and recruit another protein that physically interacts with the activation domain to enable suppression.
2. Line 58-59: “DYRK1A could enhance the mRNA level of Tau in a kinase-independent manner [25]”. The authors concluded in reference [25] that “Dyrk1A did not enhance tau gene transcription, but increased tau mRNA stability”. This should be clarified in the introduction as the authors in the current manuscript argued that Dyrk1A can enhance gene transcription. These two distinct mechanisms should be further discussed in Section 4.
Author Response
Reviewer 1
Quyang et al., presented clear evidence in this manuscript to support the transcriptional function of CrDYRK1 that is independent of its kinase activity. The authors also identified the C-terminal activation domain and the N-terminal repressor domain related to the transcriptional function of CrDYRK1 and that this on-off mechanism is conserved in human DYRK1A but not DYRK1B. To explain the repression mechanism, the authors provided subcellular localization evidence to argue that the repressor domain enables extra nuclear localization of cDYRK1. However, a major concern is that only a small fraction of cytoplasmic and membranous localization cannot explain a complete loss of transcriptional activation for isoform II. Using RNA-seq and bioinformatics analyses, the paper provided potential transcriptional targets and pathways regulated by human hDYRK1A that is related to its role in neurological functions. Overall, the manuscript is well written. English editing is highly recommended to enhance readability. Before this paper is considered for publication, the following questions must be addressed by either experimentation or more insightful discussion. The manuscript is in a good shape for publication, I would request the author to address the following comments:
Major Points:
- A detailed description of the transcription activation assay must be inserted in section 3.2 to fully explain the data.
Response: Thanks for your kind suggestion. We have provided the detailed description of the transcription activation assay in the revision (Line 276-289), which could help the audience to well understand our experimental designs and results.
- Supplemental Figure 1B must be inserted as a figure in Section 3.4 to support the point that residues of 1-168 of isoform II confers cytoplasmic and membranous localization. In this figure, co-localization of GFP-cDYRK1 with nuclear stain by DAPI and a membrane marker would significantly enhance the conclusion. Also, showing that GFP-cDYRK1 (isoform II) (85-815) loses its cytoplasmic and membranous localization is required to add further evidence to explain why this truncated protein acquire the ability for transcriptional activation, as shown in Fig. 2F. Is this also true for human DYRK1A? Is the presence of the N-terminal fragment 1-69 sufficient to relocate hDYRK1A (485-763) from the nucleus to the cytoplasm and membrane?
Response: Thanks for the insightful comment. We recognized that the evidences in Supplemental Figure 1 was not sufficient to elucidate the suppression mechanism of fragment 1-69. They are pieces of cues and more speculative, and we think it is not appropriate to have these results as a separated section. In the contrast, we have merged the Section 3.4 with the Section 3.3 together (Line 350-360), which will not affect our major conclusion.
In addition, we provided the DAPI staining in Supplemental Figure 1B for showing the nuclear localization of GFP-cDYRK1. The phalloidin stained actin filaments and could depict the cell shape. The merge of GFP-cDYRK1 signaling with phalloidin staining at the boundary indicates the potential cell membrane localization of GFP-cDYRK1. Unfortunately, we could not perform additional experiments to provide the localization of GFP-cDYRK1 (isoform II) (85-815) and hDYRK1A (485-763) due to close of lab at the period of pandemic Covid-19 infection and Chinese Spring Festival holiday. Considering the comment 2 and 3 from this reviewer, we agree that we could not have a conclusion on the inhibition mechanism even with these data. The predicted protein structures and change of subcellular localization are just cues for the speculation. Therefore, we merged two sections and toned down our statement in the revision.
- GFP-cDYRK1 (isoform II) still exhibited strong nuclear localization (supplemental figure 1b) yet it had little transcription activation activity, as compared with GFP-cDYRK1 (isoform I) (Fig. 2B). Therefore, it is impossible to explain the complete lack of transcription activity by only a small fraction of GFP-cDYRK1 (isoform II) outside the nucleus.
Response: We are agree with the opinion of this reviewer and recognize that changes of subcellular localization of the CrDYRK1 (isoform II)::eGFP construct is not adequate at all to interpret the inhibition mechanism. The differential localization of CrDYRK1 isoforms might be one of the potential possibilities for repressing the tans-activation activity. However, we could not rule out the other possibilities based on our current data. For example, the N-terminal fragment could recruit some transcription repressors to perform inhibit function. In the future, we will address this interesting inhibition mechanism for the transcription activity.
Minor Comments:
- Line 255-258: “This indicated the suppression effect might not rely on the physically bound between repression domain and activation domain (Supplementary Figure 1A)”. The ALPHAFOLD prediction implicates that the suppression effect might not reply on intramolecular interaction between the repression domain and the activation domain. It is still possible the repression domain can bind and recruit another protein that physically interacts with the activation domain to enable suppression.
Response: We agree with the insightful comment from this reviewer. Indeed, the repression domain might perform function by recruiting and binding another trans-activation repressors. We have modified the text and added the word “intramolecular” into the sentence (Line 353).
- Line 58-59: “DYRK1A could enhance the mRNA level of Tau in a kinase-independent manner [25]”. The authors concluded in reference [25] that “Dyrk1A did not enhance tau gene transcription, but increased tau mRNA stability”. This should be clarified in the introduction as the authors in the current manuscript argued that Dyrk1A can enhance gene transcription. These two distinct mechanisms should be further discussed in Section 4.
Response: Thanks for pointing our mistake. We revised the description in the revision (Line 60-62) and discussed these two distinct mechanisms in Section 4 (Line 604-610).

Reviewer 2 Report
The study described in the submitted manuscript from Ouyang et al. focuses on DYRK1 in Ciona robusta, an important model system for developmental and evolutionary biology research. The authors first showed that the kinase domain in DYRK1 is highly conserved across species. By mapping the domains in different isoforms of CrDYRK1, HsDYRK1A and HsDYRK1B, the authors demonstrated that: 1, the C-terminal of CrDYRK1 showed transcriptional activity; 2, the N-terminal domain autoinhibited the transcriptional activity. The authors next performed the RNA-seq assay from the Hela cells expressing HsDYRK1A and revealed its potential roles of transactivation ability. This is a competently-performed study, rich in data, which examines the function of DYRK1 from various angles. It will clearly be of interest to researchers who study DYRK1 and also to the more general dev-cell biology audience. There are, however, some issues, detailed below, which first require attention from the authors. In particular, the Discussion needs to be modified to focus on discussing the results.
In particular, I have the following major concerns.
1. The authors summarized the functions of DYRK1A and DYRK1B with proper literature citations in the second paragraph (Lines 51-62) of the introduction. However, it confused me since the authors used DYRK1 in some sentences. The authors should clarify DYRK1A or DYRK1B based on their citations.
2. The authors used a paragraph (Lines 63-68) to present the advantage of Ciona to motivate their study in the introduction. This part was a bit lengthy. I would suggest shortening this part to 1-2 sentences and combining it with the last paragraph of the introduction. Further, the authors could discuss the advantages of Ciona in this study in their Discussion.
3. There should be a section for the detail of staining and imaging in the Materials and Methods. In this section, the information on Phalloidin, DAPI, any other antibodies (if used) and the objective for imaging should be presented.
4. In sections 3.2, 3.3, 3.4, 3.5 and the first paragraph of 3.6. The data should be distributed more if the text has no character limit. For example, “Then, we overexpressed CrDYRK1 (isoform I)::eGFP and CrDYRK1 (isoform II)::eGFP in Ciona notochord cells driven by Brachyury promoter. The results showed that CrDYRK1 (isoform I) and CrDYRK1 (isoform II) had different subcellular localization.” (Line 261-263) could be “ Then, we expressed CrDYRK1 (isoform I)::eGFP and CrDYRK1 (isoform II)::eGFP in Ciona notochord cells driven by Brachyury promoter. The GFP signal was detected at the nuclear from the isoform I expressing cells. Meanwhile, the GFP signal was detected at the nuclear, cytoplasm and cell members from the isoform II expressing cells.”
5. The sub-cellular localization, particularly in the nuclear, of CrDYRK1 was described in Lines 261-271 and Figure S1B. There could be images showing the overlapping of eGFP signal and nuclear markers, such as DAPI staining.
6. In Lines 374-351 and Figure 5, the authors mentioned the “risk mutations”. Can the author explain how they defined the “risk mutations”?
7. The scale bars in Figure S1B and Figure S3 could be more obvious than the current ones.
8. The author mentioned that “The red underline indicates the homologous region.” In Figure S2. I wonder if “The red underline” is correct in the figure.
Minor comments:
1. Line 48. It should be “expression of DYRK1 might be regulated…”
2. Lines 175-176. It should be “ To identify the functional conservation of DYRK1, we performed the domain analysis and found that…”
3. Line 200. It should be “ There are three isoforms of CrDYRK1 in ascidian C. robusta.”
4. Line 209. It should be “ and found that the activation…”
5. Line 255. It should be “ This suggested that the suppression effect…”
6. Line 366. It should be “To investigate the mechanism of transcription activation…”
7. Much of the written language is awkward and not grammatically correct in Discussion. Such as: “however, the work thinks that” in Line 400, “Our work found the activation activity” in Line 405, and so on. The authors should check this part carefully.
Author Response
Reviewer 2
The study described in the submitted manuscript from Ouyang et al. focuses on DYRK1 in Ciona robusta, an important model system for developmental and evolutionary biology research. The authors first showed that the kinase domain in DYRK1 is highly conserved across species. By mapping the domains in different isoforms of CrDYRK1, HsDYRK1A and HsDYRK1B, the authors demonstrated that: 1, the C-terminal of CrDYRK1 showed transcriptional activity; 2, the N-terminal domain autoinhibited the transcriptional activity. The authors next performed the RNA-seq assay from the Hela cells expressing HsDYRK1A and revealed its potential roles of transactivation ability. This is a competently-performed study, rich in data, which examines the function of DYRK1 from various angles. It will clearly be of interest to researchers who study DYRK1 and also to the more general dev-cell biology audience. There are, however, some issues, detailed below, which first require attention from the authors.
In particular, the Discussion needs to be modified to focus on discussing the results.
Major concern
- The authors summarized the functions of DYRK1A and DYRK1B with proper literature citations in the second paragraph (Lines 51-62) of the introduction. However, it confused me since the authors used DYRK1 in some sentences. The authors should clarify DYRK1A or DYRK1B based on their citations.
Response: Thanks for your kind suggestion. We apologized for the confusion and now clarified DYRK1A or DYRK1B (Lines 53-64) based on the description in citations.
- The authors used a paragraph (Lines 63-68) to present the advantage of Ciona to motivate their study in the introduction. This part was a bit lengthy. I would suggest shortening this part to 1-2 sentences and combining it with the last paragraph of the introduction. Further, the authors could discuss the advantages of Ciona in this study in their Discussion.
Response: Thanks for your suggestion. We have shortened and merged this part with the previous paragraph in the Introduction (Lines 65-67). The part of the advantages of using Ciona model has been moved to the Discussion section (Lines 586-591).
- There should be a section for the detail of staining and imaging in the Materials and Methods. In this section, the information on Phalloidin, DAPI, any other antibodies (if used) and the objective for imaging should be presented.
Response: Thanks for your kind suggestion. We provided the details of staining and imaging in sections 2.7 of the Materials and Methods (Lines 214-228). We also provided the information regarding Phalloidin, DAPI, and the objective for imaging in sections 2.7 (Lines 218-221).
- In sections 3.2, 3.3, 3.4, 3.5 and the first paragraph of 3.6. The data should be distributed more if the text has no character limit. For example, “Then, we overexpressed CrDYRK1 (isoform I)::eGFP and CrDYRK1 (isoform II)::eGFP in Ciona notochord cells driven by Brachyury promoter. The results showed that CrDYRK1 (isoform I) and CrDYRK1 (isoform II) had different subcellular localization.” (Line 261-263) could be “Then, we expressed CrDYRK1 (isoform I)::eGFP and CrDYRK1 (isoform II)::eGFP in Ciona notochord cells driven by Brachyury promoter. The GFP signal was detected at the nuclear from the isoform I expressing cells. Meanwhile, the GFP signal was detected at the nuclear, cytoplasm and cell members from the isoform II expressing cells.”
Response: Thanks for your suggestion. We have made modifications on the text of these sections.
- The sub-cellular localization, particularly in the nuclear, of CrDYRK1 was described in Lines 261-271 and Figure S1B. There could be images showing the overlapping of eGFP signal and nuclear markers, such as DAPI staining.
Response: Yes. We have provided the DAPI staining in Figure S1B to show the colocalization of DAPI and GFP-CrDYRK1.
- In Lines 374-351 and Figure 5, the authors mentioned the “risk mutations”. Can the author explain how they defined the “risk mutations”?
Response: We apologized for the confusion due to our unprecise description. These mutations are derived from clinical genomic data and potentially correlated to DYRK1A-related intellectual disability syndrome in the description of UniProt database. We have replaced the “Risk mutation” with “Mutation associated DYRK1A-related intellectual disability syndrome” in Figure 5 and revised the text for the description of these mutation in the revision (lines 17, 536, 543-544). The exact effects of these site mutations are unknown.
- The scale bars in Figure S1B and Figure S3 could be more obvious than the current ones.
Response: We have made the bar more visualizable in in Figure S1B and Figure S3.
- The author mentioned that “The red underline indicates the homologous region.” In Figure S2. I wonder if “The red underline” is correct in the figure.
Response: Sorry for the mistake. We replaced “The red underline” with “The red line” in the revision.
Minor comments:
- Line 48. It should be “expression of DYRK1 might be regulated…”
- Lines 175-176. It should be “To identify the functional conservation of DYRK1, we performed the domain analysis and found that…”
- Line 200. It should be “There are three isoforms of CrDYRK1 in ascidian C. robusta.”
- Line 209. It should be “and found that the activation…”
- Line 255. It should be “This suggested that the suppression effect…”
- Line 366. It should be “To investigate the mechanism of transcription activation…”
Response: Thanks very much for above kind corrections.
- Much of the written language is awkward and not grammatically correct in Discussion. Such as: “however, the work thinks that” in Line 400, “Our work found the activation activity” in Line 405, and so on. The authors should check this part carefully.
Response: Thanks for the suggestion. We have checked writing carefully on this part and made some modification in the revision.

Round 2
Reviewer 1 Report
The authors addressed my major concerns by removing some conclusions that were not well supported. Although I was disappointed by no further experimentation to shed new lights into the mechanism of transcriptional activity, the evidence for DYRK1 transcriptional function was strong and contributed significantly to a new perspective of DYRK1 function. I would therefore recommend acceptance after one minor but important revision, see below.
The readability of this manuscript can be significantly improved by professional editing service.
Author Response
Thanks for the comments and suggestions from this reviewer. The language of this manuscript have been carefully checked by professional editing service.